# Urine-Derived Stem Cells: Applications in Regenerative and Predictive Medicine

**DOI:** 10.3390/cells9030573

**Published:** 2020-02-28

**Authors:** Guida Bento, Aygul K. Shafigullina, Albert A. Rizvanov, Vilma A. Sardão, Maria Paula Macedo, Paulo J. Oliveira

**Affiliations:** 1CNC–Center for Neuroscience and Cell Biology, UC-Biotech, University of Coimbra, 3030-789 Cantanhede, Portugal; guida.bento@gmail.com (G.B.);; 2Institute of Fundamental Medicine and Biology, Kazan Federal University, 420008 Kazan, Russia; 3Faculty of Medicine and Health Sciences, University of Nottingham, Nottingham LE12 5RD, UK; 4Centro de Estudos de Doenças Crónicas (CEDOC), NOVA Medical School-FCM, Universidade Nova de Lisboa, 1169-056 Lisbon, Portugal; 5APDP-Diabetes Portugal Education and Research Center (APDP-ERC), 1250-189 Lisboa, Portugal; 6Departmento de Ciências Médicas, Instituto de Biomedicina-iBiMED, Universidade de Aveiro, Aveiro 3810-193, Portugal

**Keywords:** urine-derived stem cells, personalized medicine, regenerative medicine, induced-pluripotent stem cells

## Abstract

Despite being a biological waste, human urine contains a small population of cells with self-renewal capacity and differentiation potential into several cell types. Being derived from the convoluted tubules of nephron, renal pelvis, ureters, bladder and urethra, urine-derived stem cells (UDSC) have a similar phenotype to mesenchymal stroma cells (MSC) and can be reprogrammed into iPSC (induced pluripotent stem cells). Having simple, safer, low-cost and noninvasive collection procedures, the interest in UDSC has been growing in the last decade. With great potential in regenerative medicine applications, UDSC can also be used as biological models for pharmacology and toxicology tests. This review describes UDSC biological characteristics and differentiation potential and their possible use, including the potential of UDSC-derived iPSC to be used in drug discovery and toxicology, as well as in regenerative medicine. Being a new cellular platform amenable to noninvasive collection for disease stratification and personalized therapy could be a future application for UDSC.

## 1. Morphological Characterization and Early Applications of Urine-Derived Stem Cells

Whenever a tissue is under constant stress from various factors (physical, chemical or mechanical), increased cell proliferation, high regeneration rate of the tissue and presence of regional stem cells is often observed. The fast renewal capability of skin and mucous layer of the stomach is one such example. In this context, urine also contains toxic metabolic wastes, having high osmotic pressure and a nonphysiological pH [1], which converts it into an aggressive body fluid that dramatically changes with the type of insult. These characteristics justify why the urinary tract, as an excretory organ, reveals a high regeneration potential as well. Due to this, the search for regional tissue-specific stem cells of the urinary system gained momentum in the last decade.

The first cells from the urinary tract that were isolated, cultivated and characterized in vitro were exfoliated urinary cells from newborn children, initially described in 1972 by Sutherland and Bain [2]. Four years later, Linder described the culture of cells from the urine and bladder washings of adults [3]. Several follow-up papers reported the isolation, culture and growth properties of human urinary epithelial cells (urothelial cells) [4,5,6]. Independently, Herz et al. described the culture of urinary cells from adults [7,8]. 

The optimization of the culture conditions for epithelial cells from newborn urine, namely on plates covered by collagen-I matrix in serum-free medium consisting of a 1:1 mixture of Dulbecco’s modified Eagles’ medium (DMEM) and Ham’s F-12 medium supplemented with insulin, transferrin, selenium and hydrocortisone was described [9]. Under these conditions, epithelial cells were able to undergo five passages while retaining the original morphology. Another alternative methodology for epithelial cells isolation from four to six-week-old rat urinary bladders was suggested by Johnson et al. by performing an attachment of bladder mucosal explants to collagen-I gels and the addition of the epidermal growth factor (EGF) [10]. The cultured cells had similar characteristics to human urothelial cells, namely junctional complexes, desmosomes, stratification and apical glycocalyx, while the ability of derived cells to be serially passaged increased 100-fold. Since bladder urothelial cells are in contact with interstitial cells, Howlett et al. described the culture of isolated urothelial cells on the feeder layer of embryonic mesenchymal-derived (Swiss 3T3) cells and collagen-I matrices [11]. Using this protocol, the culture of urothelial cells using conditioned medium from 3T3 cells was not enough to support the expression of tissue-specific characteristics. This indicated that direct intercellular contacts are necessary. Moreover, such culture models simplified three-dimensional tissue-like facsimiles of bladder stroma [11]. Another important factor determining viability, growth kinetics and cell differentiation is the cell culture medium and its supplements [12]. Variations in calcium concentration may affect cell growth capabilities, since with high calcium concentrations viability of growing cultures decreases, suggesting an accelerated rate of cellular differentiation. On the other hand, cells fail to form stratified epithelium in low-calcium medium [12,13]. For maintenance of the stratified structure of urothelial cells in long-term cultivation, it is preferred to cultivate cells in collagen-covered flasks [14], although more effective results were achieved by cultivating cells on a porous collagen matrix in cell medium supplemented with fetal bovine serum (FBS), hormones and calcium [13,15].

Urothelial cells can be isolated not only by urine sedimentation, as previously performed, but also by biopsies from renal pelvis, ureter, bladder and urethra [16]. This method is effective, allowing the isolation of cells from distinct tissues and in larger quantities, in comparison with those obtained by using a sedimentation technique. Nonetheless, biopsy is an invasive procedure which can present different complications, during and after the manipulation, and should be avoided whenever possible.

An alternative method to obtain stratified urothelial cell layers was described by Gustafson et al. [17], involving the cultivation of urothelial cells on decellularized skin grafts. The advantage of this approach is its short time of cultivation, facilitation of transplantation procedure, opportunity to transplant undifferentiated cells and formation of a vascular network under a urothelium graft. As the authors suggested, this method provides autologous urothelium for reconstructive surgery in the genitourinary tract [17]. 

Immunomagnetic beads, conjugated with monoclonal antibodies to aminopeptidase M (APM, CD13) allowed to isolate proximal tubule cells and an addition of antibodies to Tamm-Horsfall glycoprotein, a specific antigen for the thick ascending limb, and the early distal convoluted tubule allowed the separation of distal tubular cells [18]. 

Isolation and cultivation of two other different cell types from human urine sediments were performed by Dörrenhaus et al. [19]. In this work, authors described two cell populations: (a) type 1 cells presented irregular contours, including spindle-like cells inside colonies, expressed cytokeratin-7 (marker of urothelial cells), were cultivated up to six passages and formed cobblestone-like cells and domes (hemicysts); (b) type 2 cells had smooth-edged contours, stained positively with carbonic anhydrase (marker of renal tubular epithelium) and subcultivation was not possible. According to the results, type 1 cells were thought to be a urothelial population, originated within the urinary tract, while type 2 were considered renal tubular cells [19]. Figure 1 summarizes potential sources, morphological and phenotypic characterizations of different types of urine-derived cells.

An alternative method for the isolation and long-term cultivation of differentiated rat urothelial cells was introduced by Zhang et al. [26]. Cells were isolated by enzymatic digestion of the urothelium and then cultivated during a period of four to five months, subcultured up to 18 passages and stained positively with antibody to cytokeratin 17 (marker of epithelial phenotype), forming stratified urothelial sheets similar to the native tissue. Following these results, Zhang and Frey discussed the use of this technique to obtain autologous grafts for bladder surgical augmentation [27]. In this surgical context, part of stomach or bowel is normally used, but since their mucosa is not specialized for a persistent contact with urine, gastrocystoplasty or entero-cystoplasty were accompanied by several complications. Since graft demucosalization usually led to its fibrosis and shrinkage, Zhang and Frey offered an original solution to the problem by using de-epithelialized bowel lined with autologous urothelium, achieving decreased graft shrinkage in experimental cystoplasty [27]. 

Another method for bladder urothelial cells isolation involves bladder lavage [28], first described by Linder [3]. The authors demonstrated that isolated cells needed feeder cells in order to establish primary cultures. If further cultured without the same feeder cell layer, high quality urothelial cells would then be obtained with a normal karyotype during 14 passages, while maintaining proliferation capacity after cryopreservation [28]. Despite their ability to grow in vitro, form colonies and proliferate, these urothelial cells are not considered stem cells. Although a great potential in the treatment of genitourinary tract diseases, they can only restore the mucous epithelium and do not have multi-differentiation potential, one of three criteria, together with clonogenicity and self-renewal, that characterize stem cells [21].

Stem-like cells were also described in the urinary system. Zhang et al. [21] described four different cell populations, including stem cells, isolated from voided urine, and their characteristics studied. Following this first work, other groups reproduced this experiment, and the existence of urine-derived stem cells (UDSC) was gradually accepted by other researchers [29,30]. Since UDSC can be obtained by noninvasive procedures, their isolation and cultivation are easy and inexpensive [21]; urine is now considered a promising source of stem cells, and several follow-up studies have been performed in recent years with refined approaches. 

Conditions of urine collection and storage that may affect the quality and quantity of UDSC have been studied. It was previously demonstrated that UDSC from 13- to 40-year-old humans have the highest rate of clonogenicity and, in order to obtain a large number of cells, catheterization, triple urine samples collection [21] or upper urinary tract urine collection [20] are recommended. Collection of urine from the upper urinary tract is an alternative solution for patients with bladder cancer, if the renal pelvis and upper part of the ureter are not affected [25]. Isolation of UDSC from fresh urine is preferable, but protocols for UDSC isolation can be performed within 24 h if urine is stored at 4 °C in a storage medium with serum; longer storage negatively influences the viability of cells [30]. Recommendation on the process of urine collection, including exclusion criteria (e.g., viral or bacterial infections and malignancies) for future regenerative medicine, have been previously published; nevertheless, the future will bring advances in the automatic recovery of cells from the urine and possibly rescuing UDSC from otherwise excluded individuals [31]. 

Subsequently to urine cell pellet seeding, cells tend to grow in colonies. Once cells achieve confluence, they should be subcultured. The average population doubling time varies between 20 and 29 h for fresh urine and between 28 to 32 h for urine preserved at 4 °C during 24 h [20,30]. Studies about urine preservation for UDSC isolation are still scarce [30], and it is an important point to be addressed, since it would bring several advantages to the process of UDSC isolation. Although values can vary according to culture conditions, Bharadwaj et al. achieved an average of 3.7 × 108 UDSC cells, at passage 5, in about 27 days of culture [20]. UDSC can be isolated from individuals from both genders and with an age range reported so far from 5 to 75 years [32]. In adult cells, telomerase activity is largely limited to stem cells and is associated with a high proliferative capacity. Bharadwaj et. al. described telomerase activity in 60% of independent UDSC samples [32], while a normal karyotype was found at least until cell passage 15 [20,21,29,30]. The Appendix A contains culture conditions, biological characteristics and differentiation of UDSC.

After UDSC isolation, questions about their phenotype, origin and differentiation potential arise. A common method to identify and characterize a stem cell phenotype is based on the analysis of cell surface markers known as clusters of differentiation (CD). For UDSC, the CD phenotype is similar to MSC, expressing CD73, CD90, CD105 and CD133, while CD markers of HSC (CD45, CD31 and CD34) are not present [21,25,32]. Besides distinct CD markers, the exact origin of UDSC can be studied through the detection of other specific markers for various cell populations of the urinary system. Detailed comparisons of the phenotype of UDSC with renal tubular cells, urothelial, endothelial, interstitial and other types of cells was already published by Manaph et al. [22]. Although most UDSC markers are similar to that of MSC, the former cells also express other markers of pluripotent cells, including POU5F1 or octamer-binding transcription factor 3/4 (Oct 3/4), a VMyc avian myelocytomatosis viral oncogene homologue (c-Myc), as well as renal markers, such as sine oculis homeobox homologue 2 (SIX2) or the neural cell adhesion molecule (NCAM) [22]. As suggested by Manaph et al., UDSCs should also be considered as renal progenitor cells due to the above-mentioned markers, in addition to other markers present. In other words, UDSC can be a source of MSC-like cells, which present some but not all markers typical of the latter cells, although presenting high expandability and high proliferative ability typical from other stem cells [33].

Despite this, the origin of UDSC remains controversial [20]. According to some authors, collected UDSC originate from the kidneys, because cells obtained from female patients who received a sex-mismatched kidney transplant contained the Y-chromosome and expressed normal kidney cell genes and protein markers common for parietal cells and podocytes from the upper urinary tract (renal pelvic and/or upper segments of transplanted ureter) [20]. Other authors suggested instead that UDSC originates from the basal layer of urothelium [34], pericytes of kidney [35] or urinary epithelium [25]. Such possible diversity of origins may result in the presence of various cell populations in the urine. Zhang et al. [21,26,27] described four distinct types of cells, with each one presenting distinct origins: (a) cobblestone-like cells, expressing uroplakin-Ia from urothelial origin; (b) spindle-shaped desmin+cells from muscle origin; (c) cells with circular morphology and expression of vWF (von Willebrand’s factor) from endothelial origin and (d) elongated and expressing c-kit cells, considered as interstitial cells. 

The number of cells and their origin is also determined by the procedure used for urine collection—more cells may be present in urine collected by using a urethral catheter due to catheter-induced injury of the bladder and urethral mucosa [30]. Most of the cells are basically quiescent until injury, and according to clinical and scientific viewpoints, cells from hematological (such as erythrocytes) and epithelial origin (renal tubular cells, transitional epithelial cells and squamous epithelial cells) can be found in urine, normally in low amounts [31]. 

Related with the immune system, another important marker that distinguishes UDSC from hematopoietic stem cells is class II HLA (human leucocytes antigen) glycoproteins, which are common for antigen-presenting cells (dendritic cells, macrophages, B-lymphocytes and some activated cells). According to numerous independent studies, human UDSC do not express HLA-DR (one of class II HLA glycoproteins) responsible for the triggering of an immune reaction [36,37,38,39,40]. Furthermore, this immunogenic HLA-DR glycoprotein does not appear during in vitro UDSC culture until passage 7, so absence of this marker is considered as the reason for the immune suppressive effect of USCs transplantations [40]. This fact highlights the fact that UDSC are an attractive cell source for therapeutic application. At the same time, there is no clear information about the expression levels of class I HLA markers, glycoproteins present in the cell membrane of almost all nucleus-containing cells, which restricts opportunities of allogenic transplantation. We would then agree with Gaignerie et al. when stating that the transplantation of UDSC should be addressed according to the specific HLA haplotypes of the donor [41]. It also needs to be stressed that implantation of UDSC does not lead to teratoma formation [21,25,32].

UDSC can differentiate into various cell lineages of the urinary system—cells with urothelial, smooth muscle, endothelial and interstitial markers expression [21,29,42], as well as in mesenchymal derivatives: osteoblasts, chondrocytes and adipocytes [42]. Under specific conditions in terms of culture induction media, Bhadawaraj et al. obtained cells from the three germ layers—ectoderm (neuronal differentiation); mesodermal (smooth muscle cells, as well as osteogenic, chondrogenic, adipogenic, myogenic and endothelial differentiation) and endoderm (urothelial cells) [32,43]. In vitro differentiation into smooth muscle-like cells was also performed by Bhadawaraj et al., who cultivated UDSC in a differentiation medium. UDSC acquired not only the phenotype but also demonstrated a contractile function similar to native smooth muscle cells [20]. Wu et al. recently showed that UDSC had better adipogenic and endothelial capacities, as well as the potential to originate new vascularization, when compared to bone marrow-derived stem cells [36].

Urothelial-differentiated UDSC express urothelial-specific genes and proteins, such as uroplakin-Ia and -III, cytokeratin (CK)-7 and CK-13. Renal differentiation of UDSC can be performed by cultivating cells in commercial kidney differentiation medium [42]. Importantly, and as described above, transplanted UDSC do not lead to teratoma formation [20] or kidney abnormalities [42].

Urothelial differentiation of UDSC can be induced by implantation of UDSC into bacterial cellulose. As a result, the formation of a multilayered urothelium that can be used as a urinary conduit for urological operations was reported [34]. Following this study, UDSC were first differentiated in vitro into urothelial cells and smooth muscle cells, seeded afterwards on 3D-porous small intestine submucosa scaffolds and transplanted into athymic mice. Later, the authors demonstrated the formation of tissue, similarly to native ureter [44]. To improve vascularization of the graft, human UDSC were infected with an adenoviral vector containing the mouse vascular endothelial growth factor (VEGF) gene, co-cultured with human umbilical vein endothelial cells in a collagen-I gel and implanted subcutaneously into athymic mice. As a result, the authors observed increased in vivo survival and myogenic differentiation of a UDSC graft. Besides increased vascularization, new nerve fibers were observed at 28 days of the experiment [44]. 

The potential of UDSC for therapeutic application in urologic operations was also shown by Bhadawaraj et al. [32]. Similarly, to the work by Wu et al., UDSC were differentiated into urothelial and smooth muscle cells in vitro and then seeded on small intestinal submucosa scaffolds, cultivated for 14 days and implanted into nude mice. Multilayered tissue-like structures were observed consisting of urothelium and smooth muscle, proving that UDSC could be a source of cells for tissue-engineered therapeutic strategies for patients with chronic bladder diseases or for those from whom adequate cells cannot be obtained by biopsy [32]. In another context, intravenous injection of UDSC for protamine/lipopolysaccharide-induced interstitial cystitis in rats led to inhibition of oxidative stress, inflammatory response and cellular apoptosis, thus providing improvement of bladder function in experimental interstitial cystitis [45]. 

Neurogenic differentiation of UDSC was demonstrated in vitro in several works [29,32]. Besides in vitro studies, Guan et al. investigated whether UDSC could serve as a potential cell source for neural tissue engineering in vivo and compared UDSC with adipose tissue-derived MSC (ADSC). When seeding UDSC on a hydrogel scaffold and transplanting them into rat brains, UDSC behaved similarly to ADSC. Cells survived in the lesion site and migrated to other brain areas, expressing neurogenic markers such as glial fibrillary acidic protein, β-III tubulin and nestin. The observations show that UDSC can differentiate into neuron-like cells in the rat brain and demonstrate promising potential for regenerative medicine [29]. 

Being easily harvested without using invasive procedures and having the capacity to differentiate into osteoblasts prompted the interest of UDSC in bone tissue engineering. Guan et al. demonstrated that USCs were able to adhere, survive, proliferate and differentiate into osteoblasts in beta-tricalcium phosphate (β-TCP), a typical scaffold used in bone tissue engineering [38]. The authors also further demonstrated that the combination of UDSC and β-TCP improved bone regeneration in rats, suggesting UDSC as an alternative source of stem cells for bone tissue engineering. A study from Gao et al. demonstrated that the age of urine donors influences the proliferation rates, senescence tendency and osteogenic differentiation capacity of UDSC [46]. According to the authors, UDSCs from younger donors are better models for basic research and may have more extensive clinical applications. Nevertheless, the authors suggested that, independently from age, UDSC have potential applications in bone tissue engineering as seed cells.

The range of UDSC differentiation capabilities made those cells a promising source of stem cells for the treatment of various diseases. An antifibrotic effect and inhibition of cellular apoptosis was demonstrated in experimental streptozocin-induced type 2 diabetes rats. Transplantation of UDSC inhibited cell apoptosis, contributing to reduced fibrosis development and improving pancreas, heart, kidney and bladder functions [43].

Compared to other MSC sources, collection of voided urine is noninvasive, thus painless, and safe for patients, which is particularly important for pediatric patients. Although UDSC are actively studied, their clinical application has not yet been appropriately tackled, although autologous urothelial cells, cultivated in vitro on acellular allogenic dermis, were already successfully used for the surgical treatment of hypospadias [47].

Advantages of UDSC isolation made them an attractive source of induced pluripotent stem cells, initially generated by Zhou et al. [48]. Induced pluripotent stem cells (iPSC) can be applied to model various human diseases “in vitro” and have already been established in different diseases: cardiac [49], endocrine [50], neural [51], muscular [52], aneuploidy diseases such as Down syndrome [53] and systemic lupus erythematousus [54], among others. 

UDSC may be applied, not only for regenerative medicine but may also be used for the diagnosis of genetic disorders, allowing to store samples of patients with inherited diseases. Since UDSC isolation from patients with spinal muscular atrophy, Duchenne muscular dystrophy, paroxysmal kinesigenic dyskinesia and Wilson’s disease were already successfully established, it is possible to obtain cell samples from those individuals in a noninvasive and simple manner and use those cells for detecting genetic alterations [55].

## 2. Generation of iPSC from Urine-Derived Stem Cells (UDSC)

In 2006, Takahashi and Yamanaka showed that adult fibroblasts can be reprogrammed by the addition of four transcription factors: Oct3/4, Sox2, c-Myc and Klf4, to generate iPSC, which behave as embryonic stem cells in terms of unlimited self-renewal capacity and ability to virtually give rise to all types of cells of the organism, excluding extra-embryonic tissues [56]. Furthermore, iPSC circumvents ethical issues and immune rejection of embryonic stem cells. iPCS are a very effective tool for research, preserving the characteristics of the original cell, such as genetic mutations and epigenetic alterations, with potential applications in personalized regenerative medicine, in vitro pharmacological tests and as disease model [57]. Commonly, the differentiation of iPSC into disease-relevant cell types is imperative, since the disease phenotype is often only shown in differentiated cells. It was already demonstrated that these differentiated cells have disease characteristics. As a summary of the several possible fields of iPSC applications, the following can be mentioned: (a) iPSC technologies allow to generate patient-specific iPSC lines and differentiate them in multiple necessary lineages; (b) as a cell model to study the pathophysiology of diseases; (c) as disease models to develop and test new therapeutic drugs and strategies, including cell-target specific treatment; (d) to study genetic and epigenetic abnormalities, splicing and post translational modifications; (e) to study gene-editing technologies for the treatment of genetically inherited diseases and (f) to use differentiated iPSC for cell and organ transplantations in the clinic. The ability to generate iPSC from autologous cells reduces ethical problems and the risk for immunological rejection (Figure 2).

Several human cell types, such as fibroblasts, bone marrow mesenchymal stroma cells, peripheral blood, keratinocytes, melanocytes, dental pulp stem cells, hepatocytes, amniotic fluid-derived cells, neural stem cells, cord blood stem cells and adipose stem cells, have been reprogrammed to iPSC [58]. UDSC are an alternative source of cells for iPSC generation, with the advantage of being collected by a noninvasive method [53,54,59]. Several protocols have been studied to efficiently and safely generate pluripotent stem cells, including nonintegrative approaches, without the requirement of feeders, serum and the oncogene c-MYC—for example, using the combination of the transcription factors OCT4, SOX2, SV40T, KLF4 and the miR-302-367 cluster or Oct4, Glis1, Klf4, Sox2, L-Myc and the miR-302 cluster, combined with a short incubation with four compounds (inhibitors of lysine-demethylase1, methyl ethyl ketone, glycogen synthase kinase 3 beta and histone deacetylase) [57,60]. Moreover, UDSC appear to be easier for the induction of a pluripotent state than skin fibroblasts, possibly because urinary cells do not require the mesenchymal-to-epithelial transition (MET). This idea is supported by the MET inductive expression profile found in urinary cells [57]. It would be interesting to evaluate whether some specific characteristics of a UDSC subpopulation—for example, regarding surface markers—influence the efficiency of iPSC generation and properties of themselves and cells obtained by differentiation. There is still a lack of work about the properties of both urine-derived iPSC and mature cells generated from them, obtained from healthy or unhealthy individuals. 

## 3. Urine-Derived Stem Cells (UDSC) in Drug Discovery Therapeutics and Toxicology

In silico, in vitro and in vivo approaches have been used in human predictive toxicology. However, due to the increasing production of new molecules with potential human applications, the currently available approaches do not respond quickly and effectively to the requirements of the industry. Human cells, with the advantages of recapitulating the precise physiologic signaling mechanisms, can be used in a personalized way and for high-throughput screenings. The unmet need for new toxicology models has been recognized by a committee from the U.S. National Academy of Sciences, which urged the need for more efficient and relevant methods to assess human risks of exposure to different chemicals and pharmaceuticals [61,62]. Improvement in models, including human-derived stem cells, 3D organotypic culture models, mathematical modeling of cellular processes and morphogenesis, represents a clear movement towards in vitro/in silico-based assessments [62]. In vivo studies are also frequently used in predictive toxicology, but due to ethical issues, the use of animals in scientific research has been restricted, highlighting the need for alternative approaches and new models. With a noninvasive, low-cost collection, simple isolation methods, efficient proliferation and multipotent differentiation potential, UDSC are an important discovery in cell therapy and tissue engineering, with possible applications in regenerative and personalized medicine [31,39,63]. Clonogenicity, self-renewal capacities, the ability to reprogram to iPSCs [57,64] and the ability to differentiate in vitro into several lineages make USDC potential biological models for in vitro predictive toxicological tests with higher human relevance. In fact, Guan and co-workers [64] observed that UDSC reprogram to iPSCs easier and quicker than fibroblasts or MSCs and maintain the original genetic characteristics. The authors also highlighted UDSC as a novel biological resource for personalized medicine and for the discovery of new drugs. However, the use of UDSC in pharmacology and toxicology tests has not been much exploited. Since iPSCs derived from UDSC maintain the original genetic characteristics of the donor and can differentiate in several lineages, those cells could be an added value in new drug developments for the correction of genetic defects (Figure 2). Furthermore, due to the characteristics of USDC above described, those cells could be a good biological model used in in vitro tests to predict toxicity in drug discovery. Nevertheless, this is an area of very large interest, and it is likely to grow in the next few years.

## 4. Use of Urine-Derived Stem Cells (UDSC) for Regenerative Medicine

### 4.1. Urological Tissue Engineering

For urological tissue regeneration purposes, bone marrow mesenchymal stroma cells do not efficiently differentiate into functional urothelial cells, while UDSC have a better performance [32]. UDSC differentiated into urothelial and smooth muscle cells were previously co-cultured in layers into specific 3D porous small intestinal submucosa scaffolds. The multilayer structure obtained was comparable to the tissue of the urinary tract, suggesting that UDSC can be used for urological tissue regeneration [44]. The treatment of stress urinary incontinence can benefit from the ability of UDSC to differentiate into endothelial, myogenic and neurogenic directions, as it was previously demonstrated. In this study, UDSC mixed with alginate microbeads, containing growth factors (VEGF, insulin-like growth factor 1, fibroblast growth factor-1, PDGF, hepatocyte growth factor and nerve growth factor) and collagen-I, were injected subcutaneously into nude mice. The injection of UDSC with growth factors stimulated resident cell survival, myogenesis, enhanced revascularization and innervation, while UDSC themselves differentiated into a myogenic lineage [55]. In a follow-up experiment, the authors modified UDSC with an adenovirus-containing human vascular endothelial growth factor 165 (VEGF165), known to improve cell survival, angiogenesis and the myogenesis of MSC [65]. Subcutaneous implantation of UDSC, overexpressing (VEGF165) or added to endothelial cells/human skeletal myoblasts (as control) suspended in collagen-I-containing gel into nude mice led to increased survival of implanted cells, efficient differentiation of cells into endothelial cells and myocytes, stimulated vascularization and regeneration of nerve fibers [55]. Both experimental studies were a good approach for the development of cell therapies for the correction of stress urinary incontinence.

Besides their use for stress urinary incontinence, Liu et al. investigated the usefulness of UDSC for urethral reconstruction. Isolated rabbit UDSC were seeded on small intestine submucosa scaffolds and transplanted to repair the urethral defect of white male rabbits. The graft, containing autologous UDSC, facilitated urothelial regeneration and decreased inflammation and post-surgical fibrosis. Transplanted UDSC differentiated into smooth muscle and urothelial cells, improving smooth muscle and vessels content [51]. 

Although UDSC present some limitations in terms of differentiation potential, a recent study demonstrated that, although those cells did not show chondrogenic differentiation capacity, the decellularized matrix deposited by them provided a 3D expansion substrate to promote senescent bone marrow stromal cells’ chondrogenic potential, which provided an additional approach for the application of UDSC in tissue engineering and regeneration [66].

Transplantation of UDSC may be applied, not only for bladder or urethra reconstruction but also for the preservation of the erectile function in a rat model of a cavernous nerve injury. The authors modeled bilateral cavernous nerve injury-induced erectile dysfunction in rats and injected UDSC into the penile cavernous bodies. UDSC were genetically modified with a pigment epithelium-derived factor. As a result, they increased the smooth muscle to collagen ratio and decreased expression of transforming growth factor-b1, and decreased cell apoptosis and fibrosis in the cavernous tissue were observed. Furthermore, the paracrine effects of transplanted UDSC led to nerve protection, prevention of erectile dysfunction and improvement of endothelial cell function [67]. 

### 4.2. Bone Regeneration

In another application, UDSC seeded onto bone tissue engineering scaffolds, such as β-tricalcium phosphate and calcium silicate incorporated into poly(lactic-co-glycolic acid), showed increased osteogenic differentiation and potential for bone regeneration when implanted in vivo (rat and mice). Bone formation probably involved the Wnt/β-catenin signaling pathway activation, a mechanism already known to be associated with bone formation [23,38]. Furthermore, silver nanoparticles, which are recognized as an excellent antimicrobial agent for use on scaffolds, and bone morphogenic protein 2 transduction also promote the osteogenic differentiation of UDSC in vitro and also in vivo [68,69]. Therefore, the use of UDSC to repair bone tissue injuries is clearly as promising as for genitourinary system regeneration. 

### 4.3. Neuroregeneration

In a neuronal context, UDSC can be reprogrammed into iPSC and differentiated into neural progenitor cells with further in vitro induction to neurons-, oligodendrocytes- and astrocytes-like cells. In the same work, the authors suggested a method of purification of neural progenitor cells from undifferentiated cells using fluorescence-activated cell sorting with the A2B5 antibody. Being transplanted into a mouse with contusion of the thoracic spinal cord-purified neural progenitor cells survived, integrated into the injured spinal cord, and differentiated into neurons and glia without tumor formation [51]. Additionally, when UDSC were injected directly into the rat brain, positive neuronal markers were found after three weeks, suggesting that the brain environment can induce neuronal differentiation of UDSC [29].

### 4.4. Dental Reconstruction Applications

Dental tissue was also previously obtained by differentiation of iPSC generated from UDSC [70]. iPSC were generated from human UDSC and were differentiated into epithelial sheets while placed on the top of one mouse dental mesenchyme (E14.5). One to two days after incubation, this chimeric culture graft was transplanted into the renal subcapsular layer of adult nude mice. Interestingly, around 30% of these transplanted sheets developed into ameloblasts in a tooth-like structure. Thus, human iPSC, pre-differentiated into epithelial sheets, may contribute to tooth development by responding to odontogenic signals from embryonic dental mesenchyme. These results may be useful for further drug screening or regenerative therapies in dentistry [70]. 

### 4.5. Urine-Derived Stem Cells (UDSC) Extracellular Vesicles and Exosomes 

Beyond UDSC themselves, it has been recently proposed that exosomes released from those cells have also therapeutic activity. Chen et al. demonstrated that systemic injection of extracellular vesicles from human-harvested UDSC prevented osteoporosis in ovariectomized mice by transferring collagen triple-helix repeat containing 1 (CTHRC1) and osteoprotegerin (OPG) [71]. The same strategy was done to improve erectile dysfunction in a diabetic rat model—this time, by means of transferring pro-angiogenic miRNA [72]—and to decrease kidney injury in a diabetic rat model by transferring pro-vascular regeneration and cell protection factors, including growth factor, transforming growth factor-β1, angiogenin and bone morphogenetic protein-7 [73]. In the same disease context, Jin et al. demonstrated that exosomes from UDSC attenuated podocyte apoptosis in a mouse model of diabetes, besides decreasing apoptosis and increasing autophagy fluxes when added to MPC5 cells exposed to hyperglycemia. The molecular mechanism proposed was through increasing the expression of miR-486 [74]. 

UDSC-derived extracellular vesicles (EVs) may also play an indirect role towards the immune system. Recently, an immunomodulatory effect of EVs from MSCs has been raised mainly due to its endocrine/paracrine effects on damaged tissues. EVs can transfer proteins, lipids, mRNAs, and long noncoding RNAs between various cell types, thus mediating intercellular communication and signaling. Indeed, depending on its origin, EVs may aggravate or promote injury, as in cancer, or they are capable of modulating immune cells [75,76,77]. To highlight this phenomenon, in a rat renal injury model, MSC-EVs were able to reduce IL1-B and TNF-alpha [76,78]. The expression of IL-10, an important anti-inflammatory cytokine, was also shown to be regulated by EVs [79]. Other authors have suggested that MSC-EVs are able to control renal macrophage infiltration [80]. The precise mechanism by which EVs regulate the immune system is still unknown. One of the possible mechanisms is through microRNAs. For example, both miRs 21 and 199a, or miR-15a, are described to play an immunomodulatory role or even modulate macrophages chemoattraction [81,82,83]. Although this has not been established yet, it is reasonable to think that UDSCs could also have an immunomodulatory role. Nevertheless, to establish this new capacity, more comprehensive research needs to be performed.

In conclusion, UDSC or their derived EVs can potentially be used in cell therapy for a variety of conditions, although more studies are needed, as well as their clinical translation discussed. Of interest is the fact that exosomes released by UDSC appear to carry biologically active molecules, with meaningful and relevant therapeutic impacts. 

## 5. Potential or Current Utilization of Urine-Derived Stem Cells as Disease Models

For the study of disease staging and drug development assays, animal models have the drawback of not always accurately mimicking the precise human gene expression and physiology, which limits extrapolating results from animal models to humans. Although human cells are certainly a good option for disease modeling, their collection is often invasive and poses ethical problems. Cell-based technologies are an opportunity to model various human diseases in a culture dish. For such purposes, differentiated cells are needed; however, their isolation and long-term cultures are still a limiting step. In this case, the use of iPSC may be a good alternative, although the approach requires characterization, including the reprogramming and in vitro differentiation processes [84]. A great advantage of iPSC is the ability of human disease-specific iPSC to present the genetic background of a targeted disease [85,86,87]. UDSC can be successfully reprogrammed into iPSC, and their ability of differentiation into multiple fates can generate models for different conditions: metabolic, autoimmune and hemorrhagic disorders and diseases of cardiovascular, nervous, muscular or urinary systems, among many others. One such example is hemophilia A, from which iPSC were previously generated from human UDSC, which were further differentiated into hepatocytes that did not produce blood coagulation factor VIII, resembling patient cells [88]. Since the isolation of UDSC is noninvasive, they are a safe and promising source of stem cells from patients with bleeding disorders, since all invasive procedures are life-threatening for them [89]. The CRISPR-Cas9 nuclease system was successfully used by Park et al. on iPSC from patients with hemophilia A caused by chromosomal inversions that involve introns 1 and 22 of the F8 (blood coagulation factor VIII) gene [90]. The CRISPR-Cas9 technology was used in iPSCs in order to revert chromosomal segments to the appropriate position, being later differentiated into endothelial cells. These endothelial cells expressed the F8 gene and functionally rescued factor VIII deficiency in a mouse model of hemophilia. 

Another example of iPSC application regards fibrodysplasia ossificans progressiva (FOP), a genetic disease caused by a heterozygous missense mutation in the activin receptor-like kinase 2 gene (ACVR1/ALK2) and characterized by progressive heterotopic ossification of soft tissues [91]. iPSC generated from urine cells from patients with fibrodysplasia ossificans progressiva showed ALK2 mutations that led to a decreased ability of cells to differentiate into bone and appeared to have reduced expression of the VEGF receptor 2 in differentiated endothelial cells. This cell line can be applied as an alternative research model to evaluate ALK2 bioactivity and therapeutic drugs to inhibit the mineralization of soft tissues [92].

It has been previously observed that iPSC derived from human UDSC can differentiate into human alveolar type II (AT II) epithelial cells, which are responsible for the production of surfactants. Differentiated cells showed outstretched and epithelium-like morphology and expressed surfactant proteins A, B and C-like mature human AT II cells, modeling lung disease caused by the dysfunction of AT II cells [93]. 

The same approach was also used to obtain iPSC from UDSC of patients with Down syndrome. These T21-iPSC differentiated into glutamatergic neurons that had the ability to fire action potentials and into cardiomyocytes that had spontaneous contractions and were sensitive to the β-adrenergic agonist isoproterenol [53]. 

The identification of cardiac dysfunction in patients with specific genetic backgrounds is problematic, since a source of human cardiomyocytes is limited. Facing this, Jouni et al. generated iPSC from UDSC of patients with a rare genetically-inherited cardiac disease, the long QT syndrome. Differentiation of the generated iPSC into functional cardiomyocytes allowed to define trafficking defects, which led to a delayed rectifier potassium current and prolonged action potential. In patients, such changes can lead to potentially fatal cardiac arrhythmias [49]. 

UDSC from healthy individuals and muscular dystrophy patients can also be directly reprogrammed, by transduction with a tamoxifen-inducible MyoD lentiviral vector into functional myogenic cells, which express transcripts for muscle markers, form sarcomeres and have the capacity of contraction while harboring the mutation [94]. Furthermore, CRISPR/Cas9-edited USC produced the most frequent mutation responsible for limb-girdle muscular dystrophy type 2C [94]. Additionally, the antisense oligonucleotide strategy was used to correct the defect of dystrophin exon 44, by exon skipping, in UDSC differentiated into myogenic cells from Duchenne muscular dystrophy patients and healthy controls [95]. 

iPSCs were also used to generate cardiomyocytes from patients with cryptorchid-specific iPSC lines with genetic variations [96], hepatocyte-like cells with mutated proprotein convertase subtilisin/kexin type 9 to mimic the pathophysiology of autosomal dominant hypercholesterolemia [97] and systemic lupus erythematosus-specific iPSC from UDSC [54]. 

## 6. Advantages and Challenges of Urine-Derived Stem Cells (UDSC)

As described above, urine collection preceding UDSC isolation is easy, safe, noninvasive and inexpensive, unlike surgical biopsies associated with the collection of other stem cell types. UDSC can be isolated from healthy and unhealthy individuals while maintaining proliferation and differentiation capacity, allowing the possibility of large-scale noninvasive sample collection and storage. Furthermore, no major ethical concerns are applied to UDSC collection, and they can be used for personalized and high-throughput research or clinical applications, such as in vitro pharmacological tests, regenerative medicine and as a disease model. 

Regarding challenges and open questions, UDSC biological characterizations need to be deeper explored, with a special focus on donor age effects on UDSC properties, as previously mentioned. Moreover, in UDSC, the expression levels of some characteristic surface markers, analyzed by flow cytometry, show discrepancy among different studies [20,21,29,32,38,46,68,69,73,84,95,98,99,100]. These discrepancies can be attributed to different culture conditions, such as the composition of the culture medium, or can be a native characteristic of the isolated cells that can be heterogeneous within the same individual or between different individuals. Further studies to gather relevant information about expression levels of surface markers and their significance for UDSC biology are important.

Conceivably, autologous UDSC can be used to mitigate age-related conditions, provided that cells are collected at a younger age and cryo-preserved. Nonetheless, the maintenance of UDSC features when isolated from old individuals is not adequately assessed and is a crucial question to unveil. Gao et al. compared UDSC isolated from different age groups (5 to 14 years old, 30 to 40 years old and 65 to 75 years old), showing that, although the older group can still proliferate and undergo osteogenic differentiation, these properties decrease with age, and the count of senescent cells increases [46].

The maintenance of UDSC properties after cryopreservation would be beneficial for some experimental designs. However, to our knowledge, this process was not well-described. Consequently, along with the impact of donor age on the properties of isolated UDSC, the origin of UDSC, urine preservation methods for subsequent UDSC isolation and cryopreservation methods maintaining UDSC properties needs to be explored. Considering that there is a negative impact of aging on UDSC biology, the discovery of approaches to improve the properties of UDSC from elder individuals has considerable relevance before clinical applications.

As reviewed by Ji et al. [84], human individuals have a diverse genetic background, thereby increasing the variability of obtained results and making data explanation difficult. To overcome this issue, it would be beneficial to enlarge the population sample or establish the control and experimental group from the same cell source with the matching genetic background. The second possibility implies that the experimental group is created by targeted genome editing. It has been reported that iPSC can retain epigenetic transcriptional memory from the donor, and these traits change according to the original cell type and can influence iPSC properties such as differentiation potential with a bias towards the cell type of origin [101]. These can affect the future applications of UDSC. Concerning clinical applications, UDSC and derived cells’ safety and effectiveness conditions to be used are not established. For example, iPSC complete differentiation must be achieved before clinical transplantation, since iPSC can form teratomas when implanted.

Although numerous interesting studies have already been performed and, clearly, iPSC generated from UDSC are a promising cell source for various laboratory, fundamental and clinical applications, there are still a lot of important points to clarify, and several steps of the procedures need to be standardized. One involves the optimization and standardization of the urine samples collection method. There are various methods, including the collection of voided urine, catheterization or the collection of upper urinary tract urine. The protocol for collection can determine the type of isolated cells and their growth and proliferation potential. A variety of methods brings the diversity of isolated cells, differentiation capabilities, theories concerning the origin of UDSC and the multiplicity of results interpretations. Additionally, strict eligibility requirements for UDSC donors should be established, while all the procedures should be carried out in an ethical way and under quality control. In the same context, a rapid method for the detection of urine samples’ contamination is required [102].

## 7. Conclusions

UDCs’ impact on personalized and regenerative medicine has been increasing, despite the still-open questions. One advantage related to other sources of human biological material is the noninvasive nature of their collection, which simplifies ethical concerns. When searching Pubmed with the keywords “urine-derived cells”, we obtain a total of three publications in 2010 and 52 in 2019. It is very likely that the number will steadily increase as more is discovered about this very particular group of cells. The next years will tell whether UDC will be a critical tool in biomedicine.

## Figures and Tables

**Figure 1 cells-09-00573-f001:**
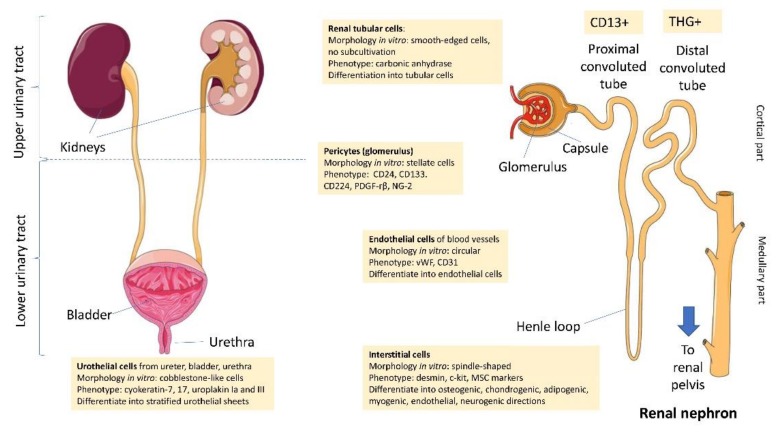
Potential sources, morphological and phenotypic characterizations of different types of urine-derived stem cells. Upper urinary tract cells can be derived from renal cortical nephron part–renal tubular cells from proximal and distal convoluted tubules [19,20], glomerular pericytes [21,22], renal interstitial cells [21,23,24] and vascular endothelial cells [21,25]. Lower urinary tract urine-derived stem cells (UDSC) belong to epithelial (urothelial) cells from ureter, urinary bladder and urethra [19,21,26]. Original source of UDSC determines their morphology in vitro, culture conditions, phenotype and differentiation capabilities. Image was adapted from original images from Servier Medical Art, as licensed under a Creative Commons Attribution 3.0 Unported License. Text was added to the original images.

**Figure 2 cells-09-00573-f002:**
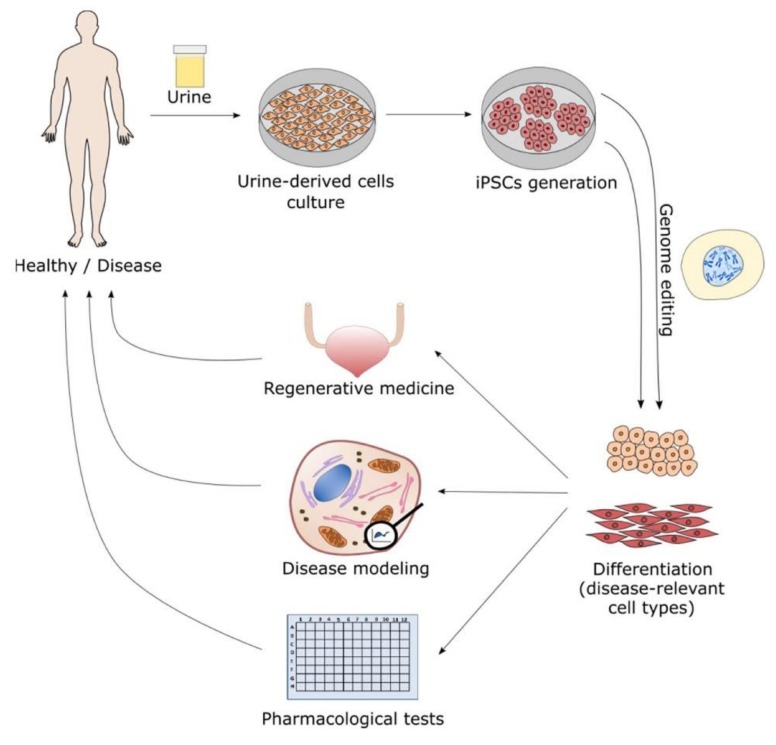
Possible applications for urine-derived cells. After urine collection from healthy or disease-affected individuals, urine-derived cells are cultured and reprogrammed into induced pluripotent stem cells (iPSC). Those cells can be differentiated into condition-relevant cell types maintaining the original genome. Otherwise, the genome can be edited to correct mutations or to introduce alterations associated with disease treatments or for research purposes. These differentiated cells can be used for autologous or allogeneic regenerative medicine as a model to study diseases or for pharmacological tests, both in personalized or high-throughput approaches.

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
