# Peer review of "Urine-Derived Stem Cells: Applications in Regenerative and Predictive Medicine"

_cells, 2020, doi:10.3390/cells9030573_

Round 1

Reviewer 1 Report

The authors summarized well about urine-derived stem cells in methodology how to collect from urine, how to induct, characteristics and quality control and how to apply in clinical and future medicine in this review article.  

Author Response

We are very thankful to the positive comments made by this reviewer, who had no concerns of other comments.

Reviewer 2 Report

The review is interesting and very rich in details that if useful for a reader with little knowledge of the importance of stem cells in regenerative medicine, it could be of little use for an expert reader. Therefore I suggest a revision of the sections in order to make reading easier for both types of readers.

The review in some sections is repetitive and full of details that may be of interest to a laboratory researcher but not very useful for the meaning of the topic for a general reader.

Here my suggestions for reordering sections and possible additions:

the figures are not original and therefore I would recommend leaving only fig. 1; I would also mention other sources of stem cells extracted from the limbus region of the eye (both for epithelial and mesenchymal limbal stem cells) (a) Simple limbal epithelial transplantation: Current status and future perspectives: Concise review. Jackson CJ, Myklebust Ernø IT, Ringstad H, Tønseth KA, Dartt DA, Utheim TP.Stem Cells Transl Med. 2019 Dec 4. doi: 10.1002/sctm.19-0203. [Epub ahead of print] Review;b )  Donor age and long-term culture do not negatively influence the stem potential of limbal fibroblast-like stem cells. Tomasello L, Musso R, Cillino G, Pitrone M, Pizzolanti G, Coppola A, Arancio W, Di Cara G, Pucci-Minafra I, Cillino S, Giordano C. Stem Cell Res Ther. 2016 Jun 13;7(1):83. doi: 10.1186/s13287-016-0342-z. Erratum in: Stem Cell Res Ther. 2016;7(1):106 ).

Section 2.1 in my opinion should be entitled: Morphological characterization of urine-derived stem cells. The chronology of the discovery of stem cells is in fact less relevant than the scientific evidence on the subject.

Section 2.3 could be summarized and enriched with important details such as the ability of these cells to express class I HLA molecules and whether there exists in vitro possibility of expressing class II HLA molecules or whether this topic  has been investigated (the latter an excellent chance to also think allogeneic transplant).

From row 241 eliminate all by inserting an Experimental animal models section and a section on Cultural conditions and Biological applications into supplemental materials.

In fact, I find that these sections are of too much laboratory interest and distract from the possible  employ of urine-derived stem cells  to the interested reader.

I find section 5 and 6 very good.

Section 7 could be shortened and summarized by a more indicative title, such as for example: Potential or in progress utilization of urine-derived stem cells.

The conclusions are incisive

Finally, table 1 is too simplistic and must be reviewed in greater detail or it can be eliminated and the data entered in the supplemental section. There are many details on the expression characteristics of stem cell molecules in flow cytometry, proteomic and genomic maps according to the different tissue origins that could be of great importance (for example the expression of HLA molecules)

Author Response

The review is interesting and very rich in details that if useful for a reader with little knowledge of the importance of stem cells in regenerative medicine, it could be of little use for an expert reader. Therefore I suggest a revision of the sections in order to make reading easier for both types of readers.

The review in some sections is repetitive and full of details that may be of interest to a laboratory researcher but not very useful for the meaning of the topic for a general reader.

Answer: Thank you for the general comments. As described below, we made significant changes to the structure of the manuscript.

the figures are not original and therefore I would recommend leaving only fig. 1; I would also mention other sources of stem cells extracted from the limbus region of the eye (both for epithelial and mesenchymal limbal stem cells) (a) Simple limbal epithelial transplantation: Current status and future perspectives: Concise review. Jackson CJ, Myklebust Ernø IT, Ringstad H, Tønseth KA, Dartt DA, Utheim TP.Stem Cells Transl Med. 2019 Dec 4. doi: 10.1002/sctm.19-0203. [Epub ahead of print] Review;b ) Donor age and long-term culture do not negatively influence the stem potential of limbal fibroblast-like stem cells. Tomasello L, Musso R, Cillino G, Pitrone M, Pizzolanti G, Coppola A, Arancio W, Di Cara G, Pucci-Minafra I, Cillino S, Giordano C. Stem Cell Res Ther. 2016 Jun 13;7(1):83. doi: 10.1186/s13287-016-0342-z. Erratum in: Stem Cell Res Ther. 2016;7(1):106 ).

Answer: Thank you for the comment. Based on this and other comments and to focus the review, we removed the first general section on stem cells. Because of this we did not include the above references. We believe that by removing this section, the reader is more addressed to the main topic of the review manuscript.

Section 2.1 in my opinion should be entitled: Morphological characterization of urine-derived stem cells. The chronology of the discovery of stem cells is in fact less relevant than the scientific evidence on the subject.

Answer: We fully agree with the review. We changed the title and performed some alterations which decreased more the idea of a “chronology-driven” section. We also removed the sub-divisions inside this section (now section 1), to avoid artificial separations.

Section 2.3 could be summarized and enriched with important details such as the ability of these cells to express class I HLA molecules and whether there exists in vitro possibility of expressing class II HLA molecules or whether this topic has been investigated (the latter an excellent chance to also think allogeneic transplant).

Answer: Thank you for the comment. We added up a description on this subject (page 7/8)

From row 241 eliminate all by inserting an Experimental animal models section and a section on Cultural conditions and Biological applications into supplemental materials.

In fact, I find that these sections are of too much laboratory interest and distract from the possible employ of urine-derived stem cells to the interested reader.

Answer: We removed this more experimental section and included as supplementary material, which will be made complementary to the main manuscript.

Section 7 could be shortened and summarized by a more indicative title, such as for example: Potential or in progress utilization of urine-derived stem cells.

Answer: We shortened section 7 and altered the title according to the reviewer instructions

Finally, table 1 is too simplistic and must be reviewed in greater detail or it can be eliminated and the data entered in the supplemental section. There are many details on the expression characteristics of stem cell molecules in flow cytometry, proteomic and genomic maps according to the different tissue origins that could be of great importance (for example the expression of HLA molecules)

Answer: Because the manuscript text contains already the description of what is referred to by the reviewer, we opted instead to eliminate Table 1, as the reviewer proposed.

Reviewer 3 Report

In this review by Bento et al., following a general introduction of stem cells, the authors detail the discovery of urine-derived cells and urine-derived stem cells (UDSC), how they were isolated, and their applications. The authors describe the commonalities between UDSCs and other stem cell types, along with UDSC attributes, marker expression, etc., along with detailing their applications, differentiation utilities and potential applications in therapeutics, regenerative medicine and disease modelling.  Finally, the authors conclude with some brief advantages and challenges of UDSCs. While the overall review is interesting and will be useful for the stem cell field, there are several significant changes that should be addressed that will help bring this review up to a level needed to be satisfactory.

Major comments:

The general introduction treats “mesenchymal stem cells” as true ‘stem cells’—which is controversial and likely incorrect, most stem cell and developmental biologist would categorize them as ‘progenitor’ cells. Instead many prefer to use the term, “mesenchymal stromal cells” to avoid this confusion. Treating MSCs as bona fide stem cells, as the authors do in this introduction, is misleading as MSCs are not capable of long-term self-renewal like HSCs or ESCs/iPSCs. My suggestion would be to either completely remove this introduction, as it is largely unnecessary and can be found in any review (and beginning this review with starting with section (2), or directly describe this controversy about the nature of MSCs.

Both figures 1 & 2 are both very general, and therefore really do not provide significant insight for the reader with regards to UDSCs. Also, evidence that stem cells exist within the heart, for example, and contribute to tissue homeostasis is controversial and not well-established. The authors need to provide solid, multiple references for these claims for tissue stem cells supporting homeostasis. Or just remove these claims altogether. It is not particularly relevant to this review. Instead, a figure documenting different urine-derived or urethral-derived cells, potentially highlighting culture conditions, would be much more novel and useful than the current figures.

Mesenchymal stromal cells (MSCs) can be derived from different tissues, such as bone marrow, adipose, dental pulp, etc. Is urine another source to generate MSCs? Or, are UDSC a completely different type of progenitor cell?  This question needs to be very carefully addressed for the reader. Based on the table provided they seem to be a type of MSC.

Authors should comment on the immunomodulatory function of MSC-derived vesicles in section 6.5 (as many recent papers and reviews, suggest this is one of their main functions), and speculate as to whether UDSCs could also have this role?

The authors provide a list of 5 steps that need to be standardized for iPSC-derived UDSCs to have utility. However, many of these are completely generic and apply to all iPSCs reviews—and therefore not at all useful. For example.#2, certainly you have to validate multi-lineage differentiation, this is why you test multiple clones—And yes, it is obvious, differentiation efficiency and cell purity is important.  While I do believe #1 and #5 are important point to make, they should be integrated into a paragraph, and not as a list. The other points are generic and can be removed.

Author Response

The general introduction treats “mesenchymal stem cells” as true ‘stem cells’—which is controversial and likely incorrect, most stem cell and developmental biologist would categorize them as ‘progenitor’ cells. Instead many prefer to use the term, “mesenchymal stromal cells” to avoid this confusion. Treating MSCs as bona fide stem cells, as the authors do in this introduction, is misleading as MSCs are not capable of long-term self-renewal like HSCs or ESCs/iPSCs. My suggestion would be to either completely remove this introduction, as it is largely unnecessary and can be found in any review (and beginning this review with starting with section (2), or directly describe this controversy about the nature of MSCs.

Answer: The reviewer presents an excellent argument. In order to avoid to include ourselves in this controversy, and as described for reviewer 1, we deleted section 1, and started the manuscript by section 2.

Both figures 1 & 2 are both very general, and therefore really do not provide significant insight for the reader with regards to UDSCs. Also, evidence that stem cells exist within the heart, for example, and contribute to tissue homeostasis is controversial and not well-established. The authors need to provide solid, multiple references for these claims for tissue stem cells supporting homeostasis. Or just remove these claims altogether. It is not particularly relevant to this review. Instead, a figure documenting different urine-derived or urethral-derived cells, potentially highlighting culture conditions, would be much more novel and useful than the current figures.

Answer: Following the reviewer comments, we decided to remove Figures 1 and 2, and to make a new Figure (now #1), documenting different urine-derived or urethral-derived cells, potentially highlighting culture conditions, and with proper references in the legend.

Mesenchymal stromal cells (MSCs) can be derived from different tissues, such as bone marrow, adipose, dental pulp, etc. Is urine another source to generate MSCs? Or, are UDSC a completely different type of progenitor cell? This question needs to be very carefully addressed for the reader. Based on the table provided they seem to be a type of MSC.

Answer: This is an excellent point raised by the reviewer. There are several evidences that UDSC are similar to MSC. We have included the arguments on section 1, pages 6 and 7. We have also replaced “mesenchymal stem cells” by the term “mesenchymal stroma cells”, as suggested by the reviewer.

Authors should comment on the immunomodulatory function of MSC-derived vesicles in section 6.5 (as many recent papers and reviews, suggest this is one of their main functions), and speculate as to whether UDSCs could also have this role?

Answer: This is a particularly good point and we thank the reviewer. We added discussion on this point on Section 4.5.

The authors provide a list of 5 steps that need to be standardized for iPSC-derived UDSCs to have utility. However, many of these are completely generic and apply to all iPSCs reviews—and therefore not at all useful. For example.#2, certainly you have to validate multi-lineage differentiation, this is why you test multiple clones—And yes, it is obvious, differentiation efficiency and cell purity is important. While I do believe #1 and #5 are important point to make, they should be integrated into a paragraph, and not as a list. The other points are generic and can be removed.

Answer: Again, we believe the reviewer is right. We revised this last section and removed generic points which were unfocused and applicable to all iPSCs reviews. Points #1 and #5 were included in a paragraph.

Round 2

Reviewer 2 Report

 i APPROVE THE FINAL VERSION OF THE MANUSCRIPT

Reviewer 3 Report

The authors have done a great job in addressing my suggested changes.